# *Ircinia ramosa* Sponge Extract (iSP) Induces Apoptosis in Human Melanoma Cells and Inhibits Melanoma Cell Migration and Invasiveness

**DOI:** 10.3390/md21070371

**Published:** 2023-06-24

**Authors:** Benedetta Romano, Daniela Claudia Maresca, Fabio Somma, Peni Ahmadi, Masteria Yunovilsa Putra, Siti Irma Rahmawati, Giuseppina Chianese, Carmen Formisano, Angela Ianaro, Giuseppe Ercolano

**Affiliations:** 1Department of Pharmacy, School of Medicine and Surgery, University of Naples Federico II, 80131 Naples, Italy; benedetta.romano@unina.it (B.R.); danielaclaudia.maresca@unina.it (D.C.M.); fabi.somma@studenti.unina.it (F.S.); g.chianese@unina.it (G.C.); caformis@unina.it (C.F.); 2Research Center for Vaccine and Drug, Research Organization for Health, National Research and Innovation Agency (BRIN), JI. Raya Bogor Km. 46, Cibinong 16911, Indonesia; peni.ahmadi@brin.go.id (P.A.); masteria.yunovilsa.putra@brin.go.id (M.Y.P.); siti.irma.rahmawati@brin.go.id (S.I.R.)

**Keywords:** melanoma, *Ircinia ramosa*, marine compounds, sponges, EMT, ROS, apoptosis

## Abstract

Marine compounds represent a varied source of new drugs with potential anticancer effects. Among these, sponges, including those belonging to the Irciniidae family, have been demonstrated to exert cytotoxic effects on different human cancer cells. Here, we investigated, for the first time, the therapeutic effect of an extract (referred as iSP) from the sponge, *Ircinia ramosa* (Porifera, Dictyoceratida, and Irciniidae)*,* on A375 human melanoma cells. We found that iSP impaired A375 melanoma cells proliferation, induced cell death through caspase-dependent apoptosis and arrested cells in the G1 phase of the cell cycle, as demonstrated via both flow cytometry and qPCR analysis. The proapoptotic effect of iSP is associated with increased ROS production and mitochondrial modulation, as observed by using DCF-DHA and mitochondrial probes. In addition, we performed wound healing, invasion and clonogenic assays and found that iSP was able to restrain A375 migration, invasion and clonogenicity. Importantly, we observed that an iSP treatment modulated the expression of the EMT-associated epithelial markers, E-CAD and N-CAD, unveiling the mechanism underlying the effect of iSP in modulating A375 migration and invasion. Collectively, this study provides the first evidence to support the role of *Ircinia ramosa* sponge extracts as a potential therapeutic resource for the treatment of human melanoma.

## 1. Introduction

Melanoma is considered to be the most lethal skin cancer worldwide, and it rapidly penetrates deep layers of the skin and infiltrates contiguous tissues, leading to metastases development. Epidemiological analyses indicate that melanoma predominantly affects fair-skinned subjects, and the number of newly diagnosed cases has been estimated to increase by more than 50% by 2040 [1,2]. Melanoma is frequently defined as “the cancer that rises with the sun” since it is strongly linked with exposure to UVA and UVB radiation from the sun or tanning lamps [3]. In fact, UV radiation exerts a genotoxic effect and can damage melanocytes DNA, inducing their transformation into cancer cells [4]. More than 50% of melanoma patients harbor the BRAF^V600E^ mutation, which constitutively activates the MAPK pathway, leading to uncontrolled proliferation [5]. The use of BRAF inhibitors, such as Vemurafenib and Dabrafenib, by melanoma patients harboring the BRAF^V600E^ mutation showed a promising effect in clinical studies [6]. However, the treatment of metastatic melanoma has strongly advanced in the last decade, with the advent of immunotherapy. Combination therapy with anti-CTLA4 and anti-PD-1 (e.g., Ipilimumab and Nivolumab) has emerged as an effective therapeutic approach for patients with metastatic melanoma that is unresponsive to BRAF inhibitors [6,7]. Nevertheless, many patients show resistance to these therapies and experience tumor recurrence, with a fatal prognosis [8]. Among the molecular mechanisms involved in drug resistance, the epithelial-to-mesenchymal transition (EMT) phenomenon is emerging as one of the major contributors to mediating resistance to conventional therapies and the newest target therapies [9]. Therefore, it is important to overcome therapy resistance by finding new targeted therapy strategies and characterizing new molecules that can thwart cancer progression. In this context, the marine world represents a very rich source of compounds that have been shown, in several studies, to fight the progression of cancer cells, and they have been approved in clinics [2,7,10]. In particular, marine sponges (phylum Porifera) secrete different bioactive natural compounds, such as terpenoids and steroids, to discourage potential predators and pathogens [11]. These products have been extensively studied for drug discovery in the tumor context [12]. Different sponge extracts have been demonstrated to induce cell death by activating multiple death pathways, such as apoptosis, autophagy and ferroptosis, in different type of cancer cells, including melanoma [12,13,14]. Among these, marine sponges of the Irciniidae family are known for their antimicrobial and anti-inflammatory activities and have been demonstrated to exert cytotoxic effects on human leukemia, breast and colon cancer cells [14,15,16]. *Ircinia ramosa* (Keller, 1889) is a prolific producer of biologically active metabolites, such as alkaloids, terpenoids and sterols, that belongs to the family Irciniidae and showed strong inhibitory activity against multiple human cancer cell lines [15]. However, so far, there are no data concerning the effect of *Ircinia ramosa* extracts on human melanoma cells. Therefore, the aim of our study was to analyze, for the first time, the therapeutic potential of *Ircinia ramosa* sponge extract (iSP) in human melanoma by evaluating its ability to modulate proliferation and induce melanoma cell death. Moreover, we also investigated the antimigratory and anti-invasive effects of iSP via modulation of the EMT phenomenon. Overall, we demonstrated that iSP was able to reduce melanoma growth and, importantly, modulate melanoma cell migration and invasion, which limited the metastases’ spread, as demonstrated via multiple in vitro approaches.

## 2. Results

### 2.1. Metabolic Profile of I. ramosa Ethyl Acetate Extract

The LC-HRESIMS/MS analysis of the *Ircinia ramosa* sponge extract (iSP) led to the annotation of a panel of diverse metabolites, such as glycerides, sterols and oxysterols, ceramides, and terpenoids [17,18,19,20,21,22,23,24] (Table 1). These metabolites were detected by comparing their exact masses with those recorded in databases, e.g., the Dictionary of Natural Products (DNP) and SciFinder based on their fragmentation pattern in the MSMS spectra.

Marine sponges belonging the genus *Ircinia* are known to be a very rich source of terpenoids, which have been found to possess diverse bioactivities [14,15,16]. Hahn et al. 2014 [17] reported the isolation of PPARδ agonistic monocyclofarnesol-derived sesquiterpenoids and their hydroxylated analogues from *Ircinia* sp., including compound 1. This latter compound has been annotated based on its fragment ions observed in the MSMS spectra at m/z 191.11, corresponding to the loss of the dimethyl-3-methylene-cyclohexane moiety and its protonated molecular ion at m/z 123.04. Above all, the metabolic profiling of the ethyl acetate extract of *I. ramosa* showed the prevalence of lipophilic compounds (2–10) that are characteristic of marine sponges.

### 2.2. iSP Treatment Affected the Proliferation Rate of Human Melanoma Cells

The antiproliferative effect of iSP on different human melanoma cell lines was assessed via MTT and compared to that on normal human epidermal melanocytes (NHEM). Cells were treated with increasing concentrations of iSP extract (0.1–100 µg/mL) for 48 h prior to evaluate cell proliferation. As shown in Figure 1A, iSP markedly reduced the cell proliferation of all melanoma cell line tested. In fact, the IC50 analysis (Figure 1B) showed a value of 199.9 µg/mL (±3.12) for NHEM, whilst for melanoma cancer cells, it was between 20–80 µg/mL, suggesting that the antiproliferative effect of iSP was specific to cancer cells. Since MTT assay is an indirect analytical method to evaluate cell proliferation, we confirmed the antiproliferative effect of iSP by performing a carboxyfluorescein succinimidyl ester (CFSE) assay [16]. In particular, A375 cells were selected, given their lower IC50 value (26.08 ± 5.47) and their aggressive phenotype [25]. As expected, the mean fluorescence intensity (MFI) of CFSE in A375 control cells at 48 h was strongly reduced compared to that of the control cells at t0, confirming the high proliferation ability of the A375 melanoma cell line. However, the treatment with iSP 30 µg/mL significantly increased the MFI of CFSE, which means a reduced proliferation rate of the A375 cells (Figure 1C,D). The antiproliferative effect of iSP was further investigated by evaluating the expression of Ki67, a key marker related to cell proliferation activity, disease progression and cancer recurrence in melanoma [26]. iSP-treated A375 cells (30 µg/mL for 48 h) showed the reduced expression of intracellular levels of Ki67 compared to those of the control (Figure 1E,F). These results demonstrated that iSP affected the proliferation rate of human melanoma cells in a selective manner, since the proliferation of normal melanocytes was not affected at the same concentration used.

### 2.3. iSP Promotes Apoptosis in A375 Human Melanoma Cells

Next, we determined whether the inhibition of cell proliferation by iSP was associated with apoptosis or necrosis. First, we assessed the live vs. dead status of the iSP-treated A375 cells using the fluorescent marker, Zombie Green. As shown in Figure 2A,B, iSP 30 µg/mL significantly increased the percentage of dead cells in A375 after the 48 h treatment compared to that of the control cells. To characterize whether cell death was associated with apoptosis and/or necrosis, we performed flow cytometry analysis via annexin V-PI double staining. We observed that iSP-treated A375 (30 µg/mL for 48 h) markedly increased the percentage of annexin V+/PI+ cells (Figure 2C). In particular, the frequency of apoptotic cells in iSP-treated A375 was 40% higher than that of the A375 control cells (Figure 2D). Likewise, the proapoptotic effect of iSP has been also tested in WM983B, another metastatic melanoma cell line, showing the same effect as that observed for A375 melanoma cells (Appendix A). This effect was further corroborated by the activation of both Caspase 9 and 3, which was evaluated using a colorimetric assay on iSP-treated A375 cells (Figure 2E). In addition, the iSP treatment markedly reduced the mRNA expression levels of two important antiapoptotic genes, the-chromosome-linked inhibitor of the apoptosis protein (XIAP) and B-cell lymphoma gene 2 (Bcl-2) (Figure 2F). Collectively, these results demonstrated that iSP induced apoptosis in melanoma cells via the activation of caspases and the inhibition of antiapoptotic genes.

### 2.4. iSP Induced ROS Production and Modulated Mitochondrial Fitness in Melanoma Cells

It is known that high levels of reactive oxygen species (ROS) induce damage to lipids, membranes, proteins and nucleic acids, leading to apoptosis [27]. Therefore, we investigated whether the suppression of cell growth accompanied by apoptosis in iSP-treated A375 was linked to increased ROS levels. The treatment of A375 with iSP (30 µg/mL for 24 h) significantly increased the generation of ROS, as assessed using a dichloro-dihydro-fluorescein diacetate (DCFH-DA) assay (Figure 3A,B). Mitochondria represent the most important source of ROS [28]. Thus, we evaluated whether iSP was able to modulate the mitochondrial mass and membrane potential of A375 melanoma cells. In line with the increased generation of ROS, iSP significantly increased both the mitochondrial mass and membrane potential, as demonstrated by using Mitotracker Green and TMRM dyes (Figure 3C–F). In addition, we also measured calcium fluxes in iSP-treated A375, given the mutual interplay between calcium and ROS signaling systems [29]. Fluo-3 AM staining showed an increase in the calcium concentration in A375 cells treated with iSP (30 µg/mL for 24 h) compared to that of the control cells (Figure 3G,H). The above results suggested that iSP modulates ROS production and mitochondrial fitness, which, in turn, are linked to iSP-mediated apoptosis. The correlation between ROS production and the apoptosis of iSP-treated A375 cells has been further demonstrated by using the ROS scavenger, N-Acetyl-L-cysteine (NAC) [30]. As shown in Appendix A, NAC did not induce apoptosis in A375 cells. Conversely, the treatment of A375 with iSP after preincubation with NAC significantly reduced the apoptosis of A375 cells. This result demonstrated that the proapoptotic effect of iSP was linked to its ability to increase ROS production.

### 2.5. iSP Induced Cell Cycle Arrest in A375 Human Melanoma Cell Line

ROS has a key role in influencing the cell cycle progression of cancer cells [27]. Therefore, we evaluated whether the treatment of A375 with iSP has an impact also on cell cycle distribution. The treatment with iSP (30 µg/mL for 24 h) showed an increased percentage of A375 cells in the G1 phase (higher than 50%), while those of the S and G2 phases were markedly reduced (20% and 30%, respectively) compared to those of the control cells (Figure 4A,B). To corroborate this result, we also evaluated the expression of CDC25A, CDK1 and CCNB1, the most important cell cycle regulatory proteins involved in cancer progression [7,31]. qPCR analysis demonstrated that the mRNA levels of CDC25A, CDK1 and CCNB1 are significantly lower in the A375 cells treated with iSP compared to those of the control cells. This result, further support the ability of iSP to arrest the cell cycle of A375 melanoma cells.

### 2.6. iSP Inhibited Melanoma Cell Migration, Colony Formation and Invasion

To evaluate whether iSP affects the migration and invasion properties of melanoma cells, we performed wound healing, colony formation and invasion assays. To avoid the fact that the effect on migration and invasion of iSP-treated A375 could be related to antiproliferative and proapoptotic effects, we decided to use the concentration of 1 µg/mL, which is very different from the IC50 value determined above. The treatment with iSP 1 μg/mL for 24 and 48 h significantly reduced A375 cells migration in a time-dependent manner (Figure 5A,B). Likewise, the colony formation assay demonstrated that iSP significantly impaired the clonogenicity of A375 cells (Figure 5C,D). In addition, the invasion assay showed that iSP significantly reduced the invasiveness of A375 melanoma cells (Figure 5E,F). All together, these results emphasized the ability of iSP to thwart the main metastatic features of cancer cells.

### 2.7. iSP Modulated the Expression of Cadherins and the EMT-Related Transcription Factors

Migration and invasion are key events involved in the epithelial-to-mesenchymal transcription (EMT) phenomenon [32]. In this context, a tumor suppressor role has been demonstrated for the epithelial protein, E-cadherin (E-CAD), in different types of cancer, including melanoma [33]. In fact, the activation of E-CAD has been demonstrated to inhibit cancer cells growth and their invasive and metastatic phenotypes [33]. Conversely, the aberrant expression of N-cadherin (N-CAD) has been found in many cancers, promoting cancer cell invasion and metastasis development [34]. Thus, we investigate whether the effects of iSP on melanoma cells migration and invasion were associated with the modulation of both N-CAD and E-CAD. Interestingly, flow cytometry analysis demonstrated that the iSP treatment (1 μg/mL for 48 h) increased the expression of E-CAD, on the one hand, and reduced the expression of N-CAD, on the other hand, in A375 cells (Figure 6A–D). Cadherins expression is regulated by different transcription factors, such as SNAIL1, SLUG, TWIST and ZEB1. These transcription factors are upregulated in melanoma cells, supporting their migration and invasion [35]. As shown in Figure 6E, iSP significantly reduced the mRNA expression levels of SLUG, ZEB1, SNAIL1 and TWIST in A375 melanoma cells. These data further demonstrate that iSP thwarted the migration and invasion of melanoma cells via the targeting of the EMT key proteins.

## 3. Discussion

The marine world represents an endless and varied source of new drugs that can be used to target different diseases, including cancer, immune- and inflammatory-based diseases [7,36]. Multiple data reported that marine compounds from micro- and macro-organisms showed promising in vitro and in vivo activities against melanoma [37]. Among these, Salinosporamide A (under the commercial name of Marizomib), a marine bacterium of the genus *Salinospora*, has been demonstrated to impair cells growth in multiple human melanoma cell lines at different aggressive stages [38]. Importantly, this compound is currently in phase III clinical trials in both US and Europe for the treatment of lung cancer, pancreatic cancer, lymphoma and melanoma [39]. Among marine organisms that exhibit a rich source of potential anticancer candidates, sponges represent a prolific source of secondary metabolites, showing promising chemotherapeutic properties [12]. For instance, Monanchocidin-A, which is isolated from the sponge, *Monanchora,* demonstrated an interesting anticancer potential, with a peculiar selectivity against melanoma cell lines [40]. Extracts from *Haliclona (Soestella) mucosa* (Griessinger, 1971), a Mediterranean sponge, was found to induce a cytotoxic effect and apoptosis in A375 human malignant melanoma cell line when used in combination with doxorubicin [41]. Halichondrin-B is a cytotoxin isolated from the sponge, *Halichondria (Halichondria) okadai* (Kadota, 1922), and it is known for its analogue, eribulin mesylate, which was approved by FDA in 2011 for the treatment of metastatic breast cancer [42]. However, Halichondrin-B demonstrated excellent preclinical activity in vivo by using different murine xenograft models of human cancer, such as colon cancer, glioblastoma, head and neck cancer, small-cell lung cancer and melanoma models [7,43]. In the last few years, new data have emerging on the anticancer effect of extracts from marine sponges of the genus *Ircinia*. For instance, it has been recently described that *Ircinia mutans* (Wilson, 1925) extracts demonstrate potent cytotoxic effects on human lymphoblastic leukaemia, colorectal cancer and breast cancer cells thanks to the rich presence of furanosesquiterpoids and sterols [44]. Likewise, the dichloromethane–methanol extract of the Indo-Pacific marine sponge, *Ircinia ramosa*, significantly impaired the proliferation of different human cancer cell lines [15]. However, there are no data regarding the antitumor effects of *Ircinia ramosa* extracts on human melanoma. Here, we demonstrated that an *Ircinia ramosa* extract (referred as iSP) impaired A375 melanoma cells proliferation, induced cell death via caspase-dependent apoptosis and arrested cells in the G1 phase of the cell cycle. Typically, cancer cells present high basal levels of ROS that support their growth and migration. Nevertheless, excessive levels of ROS evoke cancer cell apoptosis as a consequence of the damage caused to proteins, nucleic acids and lipids [27]. Several studies have shown that the effects of anticancer therapies, including immunotherapy, are related to their ability to induce excessive ROS production, resulting in cancer cell death [15]. Our results show that iSP increased ROS production in A375 melanoma cells, suggesting that iSP induces apoptosis via ROS generation. This result was further demonstrated by using the well-known ROS scavenger, NAC [30]. This finding is in line with the overwhelming list of marine drugs reported to exert antitumoral effects via ROS-mediated apoptosis, including *Ircinia* sponge extracts [45]. Mitochondria represent a major source of ROS, which are positively correlated to increased mitochondrial activity [46]. Accordingly, in our setting of iSP-induced ROS production, we observed increased mitochondrial activity that was accompanied to increased calcium fluxes, which are known to trigger ROS production [29]. Although several drugs have been developed for the treatment of melanoma in the past 10 years, resistance to therapy remains a troubling problem, resulting in relapse and the aggressive progression of melanoma within a few months [47,48,49]. Therefore, several research groups have hypothesized that targeting different signaling processes involved in cell invasion and migration could be a promising option to prevent melanoma progression. In this scenario, marine-derived compounds rank among the most promising compounds that are able to thwart cells migration and invasion in different types of cancer, including melanoma [50]. In fact, multiple marine compounds, such as Exopolysaccharide 11, Laminaran sulfate and other sulfated polysaccharides, have been demonstrated to inhibit the metastatic features of melanoma via both in vitro and in vivo approaches [51,52,53]. Consistent with the findings described above, our study showed that treatment of A375 melanoma cells with a small dose of iSP was able to restrain their invasion, migration and clonogenicity. Importantly, we also identified the mechanism underlying the antimetastatic effect of iSP in vitro. In particular, we found that the iSP treatment increased the expression of the EMT-associated epithelial marker, E-CAD, on the one hand, and reduced the expression of N-CAD, on the other hand. Both E-CAD and N-CAD are two EMT key proteins associated with therapy resistance and melanoma progression, representing very interesting targets that could be exploited for the treatment of melanoma [34,35,54]. Collectively, our results demonstrate that iSP inhibits the proliferation, migration and invasion of melanoma cells via modulating ROS production and the expression of EMT-related factors. This study provides the first evidence to support the role of *Ircinia ramosa* sponge extracts as a potential therapeutic resource for the treatment of metastatic melanoma. However, translational in vivo studies are needed to properly characterize the antitumoral and antimetastatic effects of iSP using murine models of cutaneous and metastatic melanoma.

## 4. Materials and Methods

### 4.1. Animal Material and Extraction

A specimen of *Ircinia ramosa* (Keller, 1889) (Porifera, Dictyoceratida, and Irciniidae) was collected from Kepulauan Seribu, DKI Jakarta (Indonesia), by diving at 5–10 m depths on 24 June 2022. A voucher sample (MAN/PA13-12) was deposited at the Research Center for Vaccine and Drug, the Research Organization for Health and National Research and Innovation Agency (BRIN). After homogenization, the organism (152.22 g wet weight) was exhaustively extracted using acetone three times, and then filtered. The combined filtrate was evaporated to give an acetone crude extract, which was partitioned between ethyl acetate (EtOAc) and H_2_O to obtain an organic phase (1.04 g), which was then submitted to bioactivity evaluation.

### 4.2. Chemical Analysis

The qualitative profile of the ethyl acetate extract of *I. ramosa* was performed using high HPLC-MS analysis using an Thermo LTQ Orbitrap XL mass spectrometer (Thermo Fisher Scientific Spa, Rodano, Italy) equipped with electrospray ion (ESI) MAX source coupled to a Thermo U3000 HPLC system (Agilent Technology, Milano, Italy). Chromatografic separation was achieved using a Kinetex Polar C18 column (100 × 3.0 mm, 100 Å, 2.6 µm). The injection volume was 5 µL, the flow rate 0.5 mL/ min and the mobile phase consisted of a combination of A (0.1% formic acid in water, *v*/*v*) and B (MeCN). Chromatographic analyses were conducted using a linear gradient from 50% to 95% B for 20 min, and then held at 95% B for 10 min. HRMS and MSn spectra, in the positive mode, were recorded in data-dependent acquisition mode, inducing the fragmentation of the 5 most intense peaks for each scan. Source conditions were as follows: spray voltage, 4.8 kV; capillary voltage, 31 V; auxiliary gas, 15 (arbitrary units); sheath gas, 32; capillary temperature, 285 °C; normalized collision energy, 30; isolation width, 2.0; activation Q, 0.250; activation time, 30 ms. The acquisition range was 150–1500 m/z.

### 4.3. Cell Culture 

Normal human epidermal melanocytes (NHEM) were purchased from Lonza (Walkersville, MD, USA) and were cultured in melanocyte growth medium 2 (Lonza). The A375 human melanoma cell lines were purchased from Sigma–Aldrich (Milan, Italy) and were cultured in Dulbecco’s modified Eagle’s medium (DMEM) supplemented with 10% fetal bovine serum, 2 mmol/L L-glutamine, 100 U/mL penicillin, 100 µg/mL streptomycin and 10 mM HEPES buffer (all from Gibco; New York, NY, USA). WM983A and WM983B cell lines were purchased from Rockland (Limerick, Ireland) and cultured in complete RPMI 1640 Medium. Cells were incubated at 37 °C in a humidified incubator under 5% CO2.

### 4.4. MTT Assay

Cell proliferation was measured using a 3-[4,5-dimethyltiazol2yl]-2,5 diphenyl tetrazolium bromide (MTT) assay as previously reported [55]. A375 human melanoma cell and NHEM were plated on 96-well plates (3 × 10^3^/well) and, the day after, were treated with different concentrations of iSP (0.01–100 µg/mL) for 48 h before adding 25 µL of MTT (Sigma, Milan, Italy) (5 mg/mL in saline). The plate was incubated for an additional 3 h at 37 °C prior to solubilize the blue formazan crystals produced with the DMSO solution. The absorbance of each well was measured at 490 nm with a microplate spectrophotometer reader (Multiskan FC, Thermo Scientific™, Waltham, MA, USA).

### 4.5. Flow Cytometry Analysis

To evaluate the cell proliferation rate, A375 human melanoma cells were labeled with 5 µM carboxyfluorescein succinimidyl ester (CFSE, Thermo Fisher Scientific, Waltham, MA, USA) and incubated for 20 min at 37 °C. Cells were directly analyzed or grown for 48 h in the presence of 30 µg/mL iSP before analyzing the fluorescence intensity via flow cytometry.

For Ki67 expression, iSP-treated A375 (30 µg/mL for 48 h) was fixed and permeabilized with Intracellular Fixation & Permeabilization Kit (eBioscience, Thermo Fisher Scientific Waltham, MA, USA). Next, staining was performed with APC antihuman Ki67 antibodies (REA183, Miltenyi, 2:50) and incubated for 30 min prior to acquisition using the flow cytometer.

For live vs. dead comparison, iSP-treated A375 (30 µg/mL for 48 h) was stained using the Zombie Green Fixable Viability Kit (BioLegend, San Diego, CA, USA) according to the manufacturer’s instructions. 

The apoptosis of iSP-treated A375 and WM983B (30 µg/mL for 48 h) was evaluated with the Annexin V-FITC Kit (BD Pharmingen, San Diego, CA, USA) according to the manufacturer’s instructions.

For the experiment with NAC, A375 cells were incubated 1 h with NAC, and then treated with iSP (30 µg/mL for 48 h). 

Reactive oxygen species production, mitochondrial activity and calcium concentrations were evaluated in iSP-treated A375 (30 µg/mL for 24 h) via staining the samples with 10 µM of DCFH-DA (D399 Thermo Fisher Waltham, MA, USA), 50 nM TMRM (T668, ThermoFisher Waltham, MA, USA), 10 nM MitoTracker Green (M7514 Thermo Fisher Waltham, MA, USA) and 5 µM Fluo-3 AM (F1241, Thermo Fisher Waltham, MA, USA). Staining was performed according to the manufacturer’s instructions.

The cell cycle distribution of iSP-treated A375 (30 µg/mL for 24 h) was assessed using the Cell Cycle Assay Solution Deep Red Kit (Dojindo, Kumamoto, Japan) and performed according to the manufacturer’s instructions.

E-Cadherin (E-CAD) and N-Cadherin (N-CAD) expression levels were evaluated in iSP-treated A375 (1 µg/mL for 48 h) via extracellular staining with APC-Cy7 antihuman CD324 (E-CAD) (67A4, Biolegend, 1:50) and FITC antihuman CD325 (N-CAD) (8C11, Biolegend, 1:50) as previously reported [56].

Samples were acquired using a BriCyte E6 flow cytometer (Mindray Medical Italy S.r.l., Milan, Italy), and data were analyzed using FlowJo software (TreeStar V.10; Carrboro, NC, USA).

### 4.6. Caspase 3/9 Activity Assay

The activation of Caspase 3 and 9 in A375 cells treated with A7 (30 µg/mL for 48 h) were assessed with Caspase 3 and 9 Activity Colorimetric Assay Kits, respectively, according to the manufacturer’s instructions (Houston, TX, USA).

### 4.7. Quantitative Real-Time PCR

A375 human melanoma cells were plated in 100 mm dishes (1 × 10^6^ cells/dish) and treated with iSP 1 or 30 µg/mL for 24 or 48 h. After incubation, total RNA was isolated using TRI-Reagent (Sigma-Aldrich, Milan, Italy) according to the manufacturer’s instructions. Afterwards, RNA was reverse transcribed into cDNA using iScript Reverse Transcription Supermix (Bio-Rad, Milan, Italy) as previously reported [57]. Quantitative real-rime PCR was performed using the CFX384 real-time PCR detection system (Bio-Rad) with the following primers:

BCL-2 (Gene ID: 596):

5′-GGTGGGGTCATGTGTGTGG-3′;

5′-CGGTTCAGGTACTCAGTCATCC-3′.

XIAP (Gene ID: 331):

5′-TATCAGACACCATATACCCGAGG-3′;

5′-TGGGGTTAGGTGAGCATAGTC-3′.

CDC25A (Gene ID: 993):

5′-CTCCTCCGAGTCAACAGATTCA-3′;

5′-CAACAGCTTCTGAGGTAGGGA-3′.

CCNB1 (Gene ID: 891): 

5′-GACCTGTGTCAGGCTTTCTCTG-3′; 

5′-GGTATTTTGGTCTGACTGCTTGC-3′.

CDK1 (Gene ID: 983): 

5′-GGAAACCAGGAAGCCTAGCATC-3′;

5′-GGATGATTCAGTGCCATTTTGCC-3′.

SNAIL (Gene ID: 6615):

5′-ACTGCAACAAGGAATACCTCAG-3′;

5′-GCACTGGTACTTCTT GACATCTG-3′.

SLUG (Gene ID: 6591):

5′-CGAACTGGACACACATACAGTG-3′;

5′-CTGAGGATCTCTGGTTGTGGT-3′.

ZEB-1 (Gene ID: 6935):

5′-TTACACCTTTGCATACAGAACCC-3′;

5′-TTTACGAT TACACCCAGACTGC-3′.

TWIST (Gene ID: 7291):

5′-GTCCGCAGTCTTACGAGGAG-3′;

5′-GCTTGAGGGTCTGAATCTTGCT-3′.

The S16 housekeeping gene was used as an internal control to normalize the Ct values using the 2-ΔCt formula.

### 4.8. Wound Healing Assay

A375 cells were seeded in a 6-well plate (1 × 10^5^ cells/well) and allowed to reach confluence. Next, a wound in the cell monolayer was affected with a 200 µL pipette tip. Before iSP (1 µg/mL) addition, images were captured with a microscope at time 0, and this was repeated after 24 and 48 h. ImageJ software with the MRI Wound Healing Tool (MRI Redmine) was used to calculate the area of the cell-free gap.

### 4.9. Clonogenic Assay

A375 cells were seeded in a 6-well plate (1 × 10^3^ cells/well) and allowed to attach overnight. The day after, cells were treated with 1 µg/mL iSP and incubated for 48 h. The fresh medium without iSP was changed every 2 days until colony formation. After 14 days, cells were washed with PBS, fixed with 4% paraformaldehyde and stained with 0.5% crystal violet. Colonies were manually counted, and images were acquired with a digital camera.

### 4.10. Invasion Assay

A375 cells (2.5 × 10^5^ cells/mL) were seeded in Boyden chambers previously coated with Matrigel (Becton Dickinson Labware, Franklin Lakes, New Jersey, USA) in the presence or absence of iSP (1 μg/mL) in serum-free DMEM. After 16 h incubation, chambers were fixed with 4% formaldehyde and permeabilized with 100% methanol for 20 min. Methanol was removed, and the chambers were stained with Giemsa for 15 min. Filters were placed on a slide and examined under a microscope. Cell invasion was determined by counting the number of cells stained on each filter in at least 4–5 randomly selected fields. The resulting data are presented as the average of the invaded cells ± SEM/microscopic field of three independent experiments.

## Figures and Tables

**Figure 1 marinedrugs-21-00371-f001:**
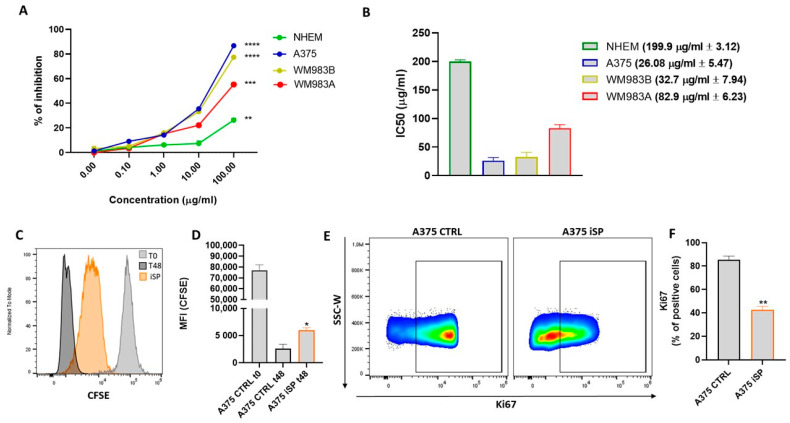
iSP affects the proliferation rate of human melanoma cell lines. (**A**) Antiproliferative effect of iSP (0.1–100 μg/mL) was assessed via MTT assay in A375, WM983A, and WM983B melanoma cells and normal human epidermal melanocytes (NHEM) at 48 h. (**B**) IC50 values for iSP-treated A375, WM983A, and WM983B melanoma cells and NHEM. (**C**) Representative example of flow cytometry analysis of CFSE staining in A375 after staining (grey histogram) and after 48 h treatment (orange histogram) or not (black histogram) with iSP 30 μg/mL. (**D**) CFSE quantification in terms of mean fluorescence intensity (MFI). (**E**) Representative example of flow cytometry analysis of A375-derived Ki67 upon treatment or not for 48 h with iSP 30 μg/mL. (**F**) Frequency of Ki67 in A375 after treatment (orange bar) or not (black bar) with iSP 30 μg/mL. Data are shown as mean ± SEM of at least three independent experiments (* *p* < 0.05; ** *p* < 0.01; *** *p* < 0.001 **** *p* < 0.0001 vs. A375 CTRL).

**Figure 2 marinedrugs-21-00371-f002:**
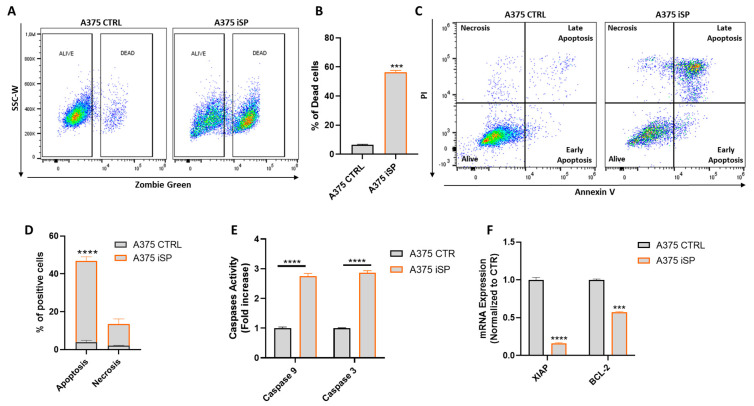
iSP induces apoptosis in A375 human melanoma cells. (**A**) Representative example of flow cytometry analysis of Zombie Green staining in A375 upon 48 h treatment or not with iSP 30 μg/mL. (**B**) Frequency of dead cells after treatment (orange bar) or not (black bar) for 48 h with iSP 30 μg/mL. (**C**) Representative example of annexin V/propidium iodide (PI) staining after 48 h treatment or not with iSP 30 μg/mL. (**D**) Frequency of apoptotic cells after treatment (orange bar) or not (black bar) for 48 h with iSP 30 μg/mL. (**E**) Activation of Caspase 9 and 3 in A375 upon 48 h treatment (orange bar) or not (black bar) for 48 h with iSP 30 μg/mL. (**F**) Expression of XIAP and BCL−2 assessed by qPCR in A375 upon 48 h treatment (orange bar) or not (black bar) with iSP 30 μg/mL. Data are shown as mean ± SEM of at least three independent experiments (*** *p* < 0.001; **** *p* < 0.0001 vs. A375 CTRL).

**Figure 3 marinedrugs-21-00371-f003:**
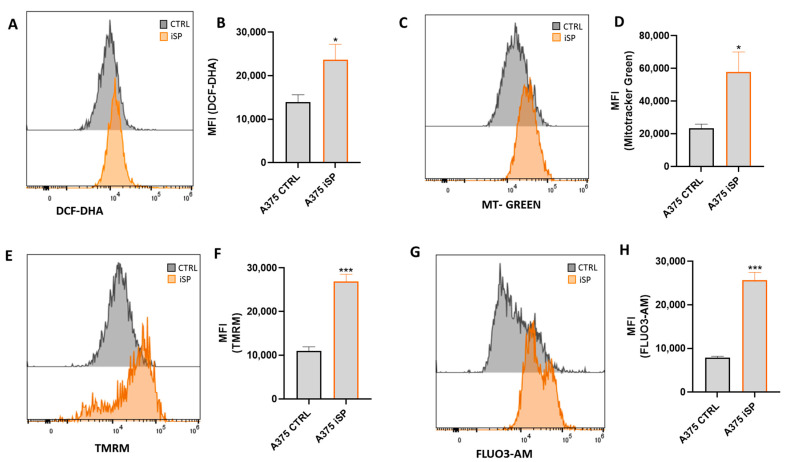
iSP induced ROS production and modulated mitochondrial fitness in melanoma cells. (**A**–**G**) Representative examples of flow cytometry analysis of DCF-DHA, (**A**) MitoTracker Green, (**C**) TMRM, (**E**) and FLUO3-AM (**G**) in untreated A375 cells (black histograms) and after iSP treatment (orange histograms) for 24 h, with their respective quantification in terms of mean fluorescence intensity (MFI) (**B**,**D**,**F**,**H**). Data are shown as mean ± SEM of at least three independent experiments (* *p* < 0.05; *** *p* < 0.001 vs. A375 CTRL).

**Figure 4 marinedrugs-21-00371-f004:**
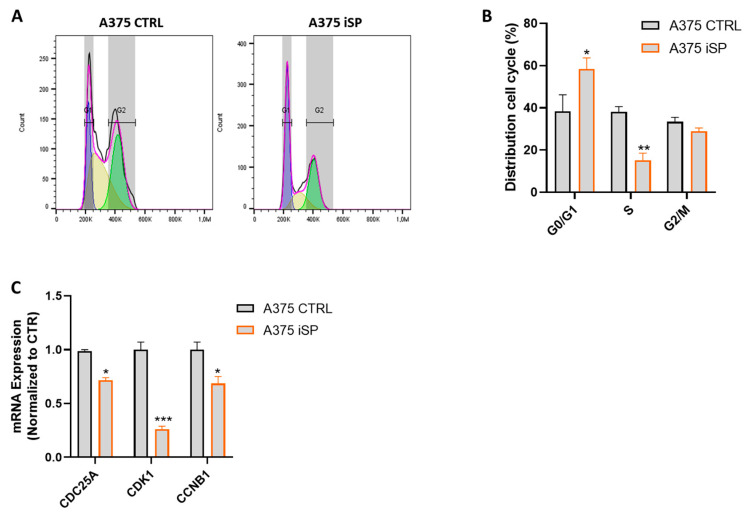
iSP induced cell cycle arrest in A375 human melanoma cell line. (**A**) Representative example of cell cycle distribution in A375 after 24 h treatment or not with iSP 30 μg/mL. (**B**) Frequency of A375 cells in G0/G1, S and G2/M cell cycle distribution in A375 after treatment (orange bar) or not (black bar) for 24 h with iSP 30 μg/mL. (**C**) Expression of CCNB1, CDK1 and CDC25C, assessed via qPCR of A375, after 24 h treatment (green bar) or not (black bar) with ERU 30 μM. Data are shown as mean ± SEM of at least three independent experiments (* *p* < 0.05; ** *p* < 0.01; *** *p* < 0.001; vs. A375 CTRL).

**Figure 5 marinedrugs-21-00371-f005:**
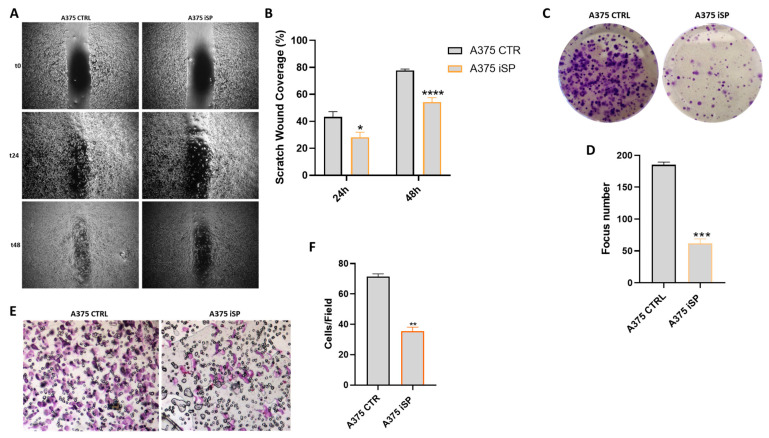
iSP inhibited melanoma cell migration and colony formation (**A**) Representative example of wound healing assay of A375 cells after incubation with iSP 1 μg/mL for 24 and 48 h. (**B**) Quantification of the healed wound area at 24 and 48 h. (**C**,**D**) Representative example (**C**) and quantification (**D**) of clonogenic assays of A375 cells after incubation with iSP 1 μg/mL. (**E**) Representative example of invasion assay of A375 cells after incubation with iSP 1 μg/mL. (**F**) Average number of invasive cells per field. Data are shown as mean ± SEM of at least three independent experiments (* *p* < 0.05; ** *p* < 0.01; *** *p* < 0.001; **** *p* < 0.0001 vs. A375 CTRL).

**Figure 6 marinedrugs-21-00371-f006:**
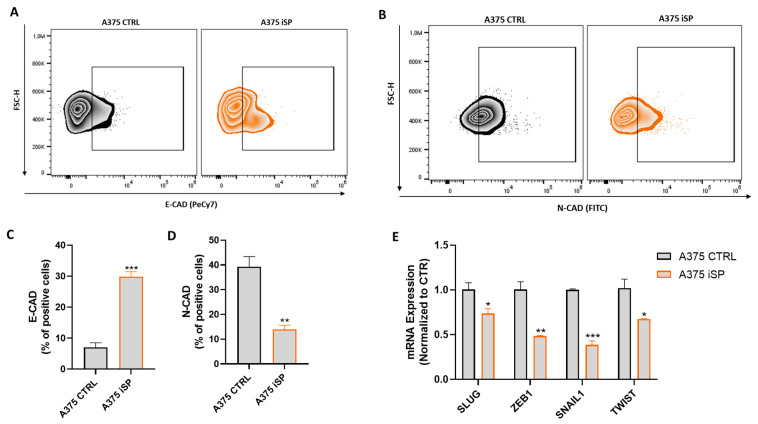
iSP modulated the expression of ECAD and EMT-related transcription factors. (**A**) Representative example of flow cytometry analysis of A375-derived E-CAD after treatment (orange dot plot) or not (black dot plot) with iSP 1 μg/mL for 24 h. (**B**) Representative example of flow cytometry analysis of A375-derived N-CAD after treatment (orange dot plot) or not (black dot plot) with iSP 1 μg/mL for 24 h. (**C**) Frequency of E-CAD in A375 after treatment (orange bar) or not (black bar) with iSP 1 μg/mL. (**D**) Frequency of N-CAD in A375 after treatment (orange bar) or not (black bar) with iSP 1 μg/mL. (**E**) Expression of SLUG, ZEB1, SNAIL and TWIST assessed by qPCR in A375 upon 48 h treatment (orange bar) or not (black bar) with iSP 1 μg/mL. Data are shown as mean ± SEM of at least three independent experiments (* *p* < 0.05; ** *p* < 0.01; *** *p* < 0.001 vs. A375 CTRL).

**Table 1 marinedrugs-21-00371-t001:** Chemical composition of *I. ramosa* organic extract. Compounds are listed in order of LC-MS elution. All mass peaks are [M + H] ^+^ adducts.

No	Family	t_R_ (min)	Measured *m/z*	Molecular Formula	Identification	Relative Amount *	References
**1**	sesquiterpenoid	1.66	315.1940	C_17_H_30_O_5_	3,4-furandiol, 3-[2-[(1S)-2,2-dimethyl-6-methylenecyclohexyl]ethyl]tetrahydro-2,5-dimethoxy	+	[17]
**2**	spongilipid	6.99	475.3234	C_25_H_48_O_9_	1-palmitoyl-3-β-D-galactosyl-glycerol	+++	[18]
**3**	acylglycerol	7.59	331.2827	C_19_H_38_O_4_	2-Hexadecanoyl glycerol	++	[19]
**4**	ceramide	12.32	368.3867	C_24_H_49_NO	Tetracosanamide	+++	[20]
**5**	N-acylated serinol	13.60	372.3451	C_22_H_45_O_3_N	Inconspicamide	++	[21]
**6**	sterol	15.12	397.3077	C_27_H_40_O_2_	Ergosterol	+	[22]
**7**	sterol	16.61	411.3235	C_28_H_42_O_2_	3β-hydroxy-24-methylcholesta-5,8,22-trien-7-one	++	[23]
**8**	sterol	16.99	381.3132	C_27_H_40_O	Cholesta-4,6,8-trien-3-one	++	[24]
**9**	sterol	18.46	383.3287	C_27_H_42_O	Cholesta-4,6-dien-3-one	+++	[21]
**10**	sterol	18.82	409.3444	C_29_H_44_O_3_	Stigmasta-4,6,8-trien-3-one	++	[21]

* Relative amount is indicated from trace level (+) to most abundant component (+++).

## Data Availability

Data are contained within the manuscript.

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
