# Peer review of "Ircinia ramosa Sponge Extract (iSP) Induces Apoptosis in Human Melanoma Cells and Inhibits Melanoma Cell Migration and Invasiveness"

_marinedrugs, 2023, doi:10.3390/md21070371_

Round 1

Reviewer 1 Report

please see attached file.

Moderate editing of English language required.

Author Response

We thank the reviewer for the interest in our work.

Please find attached the point by point reply to the comments raised.

Reviewer 2 Report

Marine natural products possess great potential for the treatment of tumors, as well as some other diseases.  As a plentiful source of secondary metabolites, sponges have contributed highly hopeful precursors for drug development, here the authors have analyzed the chemical composition of a kind of marine sponge (iSP) and first evaluated its therapeutic potential on a human melanoma. This work is therefore of general significance.

Results here demonstrate that iSP can reduce the proliferation of melanoma cells, promote apoptosis through caspases activation and inhibition of antiapoptotic genes, induce ROS production, modulate mitochondrial fitness, and importantly induce melanoma cell migration and colony formation. There is also some evidence of mechanisms underlying iSP, including modulation of expression of ECAD and EMT-related TFs. This work, therefore, has been done quite straightforwardly.  

Specific comment:

I have a concern about the concentration of iSP used in this work. The authors tested different concentrations of iSP and found 30 µg/ml can reduce the proliferation of melanoma cells significantly (Figure 1), induce apoptosis of melanoma cells (Figure 2), induce ROS production (Figure 3), and cell cycle arrest (Figure 4), but, for migration and colony formation assays of the same cells (A375), 1 µg/ml is enough, can you give an explanation? 

Author Response

(The authors gave the same response as above.)

Reviewer 3 Report

Dear Authors

The MS represents a lot of amazing work. As a marine sponge systematist I believe it is important to add in the relevant and correct classification at least in the Abstract and the first time that it appears in the body of the MS - I have made some suggestions for this

I have attached the markups on a copy of the PDF. I also think it is important to indicate permissions to collect from your partner organisation

Author Response

We thank the reviewer for the positive feedback about our manuscript. We appreciated its contribution regarding the editing of the manuscript. All the suggestions have been included in the revised version.

Reviewer 4 Report

In this manuscript, the authors suggested that iSP suppressed cell proliferation and metastatic ability of melanoma cells. Overall, the finding are interesting and novel. The diverse approach to prove their hypothesis is also impressive. However, more evidence is required to support their conclusion, and the following points should be revised.

1. The authors used only one cell line to demonstrate the anti-cancer effects of iSP. More cell lines are needed to verify whether the anti-cancer properties of iSP are consistently observed.

2. There is no evidence that iSP suppressed the invasion of A375 cells. Clonogenic assay is an in vitro cell survival assay based on the ability of a single cell to grow into a colony, not based on the invasive ability. Transwell invasion assay can be used to support the anti-invasive property of iSP.

3. In Figure 3, is ROS production responsible for the apoptosis induction? Please clarify this point by treating ROS scavenger, including NAC, in A375 cells.

4. In Figure 6, the influence of iSP on the expression of N-cadherin should be also included.

5. Please add a graphic diagram of the mechanism of anti-cancer effects of iSP in Figure 7.

Author Response

(The authors gave the same response as above.)

Round 2

Reviewer 1 Report

The manuscript may need minor English editing. 

The manuscript may need minor English editing. 

Reviewer 4 Report

Now this manuscript has been well-revised according to the reviewer's suggestion. I think it is suitable for publication.